

# Translocation of an arctic seashore plant reveals signs of maladaptation to altered climatic conditions

Maria Hällfors[1,2], Susanna Lehvävirta[2,3], Tone Aandahl[4], Iida-Maria Lehtimäki[2], Lars Ola Nilsson[4,5], Anna Ruotsalainen[6], Leif E. Schulman[2] and Marko T. Hyvärinen[2]

[1] Research Centre for Environmental Change, Organismal and Evolutionary Biology Research Programme, Faculty of Biological and Environmental Sciences, University of Helsinki, Helsinki, Finland
[2] Botany Unit, Finnish Museum of Natural History, University of Helsinki, Helsinki, Finland
[3] Department of Landscape Architecture, Planning and Management, Swedish University of Agricultural Sciences, Alnarp, Sweden
[4] Norwegian Institute of Bioeconomy Research (NIBIO), Division of Environment and Natural Resources, Ås, Norway
[5] Halmstad University, Halmstad, Sweden
[6] Department of Ecology and Genetics, University of Oulu, Oulu, Finland

Corresponding author
Maria Hällfors,
maria.hallfors@helsinki.fi

## ABSTRACT

Ongoing anthropogenic climate change alters the local climatic conditions to which species may be adapted. Information on species' climatic requirements and their intraspecific variation is necessary for predicting the effects of climate change on biodiversity. We used a climatic gradient to test whether populations of two allopatric varieties of an arctic seashore herb (*Primula nutans* ssp. *finmarchica*) show adaptation to their local climates and how a future warmer climate may affect them. Our experimental set-up combined a reciprocal translocation within the distribution range of the species with an experiment testing the performance of the sampled populations in warmer climatic conditions south of their range. We monitored survival, size, and flowering over four growing seasons as measures of performance and, thus, proxies of fitness. We found that both varieties performed better in experimental gardens towards the north. Interestingly, highest up in the north, the southern variety outperformed the northern one. Supported by weather data, this suggests that the climatic optima of both varieties have moved at least partly outside their current range. Further warming would make the current environments of both varieties even less suitable. We conclude that *Primula nutans* ssp. *finmarchica* is already suffering from adaptational lag due to climate change, and that further warming may increase this maladaptation, especially for the northern variety. The study also highlights that it is not sufficient to run only reciprocal translocation experiments. Climate change is already shifting the optimum conditions for many species and adaptation needs also to be tested outside the current range of the focal taxon in order to include both historic conditions and future conditions.

## INTRODUCTION

Ongoing climate change threatens biodiversity and is predicted, and even shown, to lead to declines in populations and to extinctions of species (*Ceballos et al., 2015*; *Dawson et al., 2011*; *Díaz et al., 2019*; *Settele et al., 2014*; *Urban, 2015*). The ability of species to respond, in situ, to shifting environmental conditions can be a result of tolerance of a broad range of conditions, or it can be mediated through evolutionary adaptation (*Chevin, Lande & Mace, 2010*; *Gao et al., 2018*; *Gienapp et al., 2008*; *Merilä & Hendry, 2014*) sometimes facilitated by gene flow from congeneric populations. In order to enable assessment of the ability of a species or population to cope with increasing pressures, it is crucial to understand through which mechanisms the impact of climate change is manifested.

Phenotypic plasticity can allow organisms to respond rapidly to changes in their environment and, hence, it has often been viewed as the main strategy to respond to environmental changes (*Merilä, 2012*). In contrast, evolutionary processes are often slow. Some species are showing signs of acclimatization or adaptation to new conditions, whereas others are either declining or dispersing to new areas with a favourable environment (*Lenoir & Svenning, 2015*; *Parmesan & Yohe, 2003*; *Pöyry et al., 2009*). To predict the effects of climate change on biodiversity and to plan effective conservation strategies, we need both evaluations of how the current distribution areas of species are changing, and approximations of the genetic and phenotypic potential of species to endure climatic changes in situ.

Species are often used as the basic taxonomic unit in plant conservation, and predictions on the effects of environmental change usually relate to the species level (*Frankham et al., 2012*). Populations of a particular species may, however, be locally adapted, and therefore have divergent affinities to abiotic conditions (*Banta et al., 2012*) and show intraspecific differences in climatic tolerance (*Atkins & Travis, 2010*; *Hill, Griffiths & Thomas, 2011*). However, in their review on local adaptation in plant species, *Leimu & Fischer (2008)* found that local adaptation of populations to environmental conditions is not as widespread as often assumed. Local adaptation may be favoured mostly in cases where populations occur in differing environments and gene flow between them is restricted (*Kawecki & Ebert, 2004*). Understanding and taking into account intraspecific variation in the adaptation to climatic conditions, is necessary to reach balanced estimates of climate change impact at the level of populations (*Hällfors et al., 2016*; *Souther & McGraw, 2011*; *Valladares et al., 2014*). The speed of climate change may outpace the ability of populations to respond adaptively, thereby causing adaptational lag (*McGraw et al., 2015*) and, hence, maladaptation instead of adaptation or acclimatization. Thus, the need to estimate the effect that climate change has had on natural populations to date, has become increasingly evident (*Anderson & Wadgymar, 2020*; *Franks, Sim & Weis, 2007*; *Kooyers et al., 2019*; *McGraw et al., 2015*; *Thomann et al., 2015*; *Wilczek et al., 2014*; *Zenni, Bailey & Simberloff, 2014*).

Manipulative experiments have the potential to provide information on how strong a role climate plays in defining favourable conditions of a species or population (*Hargreaves, Samis & Eckert, 2014*; *Kawecki & Ebert, 2004*; *Kreyling et al., 2014*; *Pelini et al., 2012*). Results from experiments where the climatic conditions for the focal taxa are altered can be incorporated

into species distribution models to improve predictions (*Greiser et al., 2020*). By reciprocally transplanting individuals of two or more populations to conditions representing a foreign environment and a home environment, we can effectively measure intraspecific variation that allows maintenance of fitness under varying temperature conditions (*Blanquart et al., 2013*; *Kawecki & Ebert, 2004*). Experiments can also test species' responses to future conditions, for example, through treatments mimicking increased temperatures, and inform us about the extent to which plasticity contributes towards population resilience under climate change. Furthermore, such experiments can reveal locally adaptive traits, which would signify that the survival of the species in the future would require either rapid microevolution *in situ*, a range shift or, in the absence of these, assisted migration (*sensu Hällfors et al., 2014*) to new favourable areas.

Here, we describe a study on Siberian primrose (*Primula nutans* ssp. *finmarchica* (Jacq.) Á. Löve & D.), in which we sampled populations of two of its varieties (the northern var. *finmarchica* and the southern var. *jokelae* L. Mäkinen & Y. Mäkinen); while acknowledging that our sampling did not cover the varieties' full distribution, for clarity we refer to the populations of the different varieties by 'variety'. We performed (1) a reciprocal transplant experiment and (2) an experiment of out-of-range transplantation. The former tests the importance of intraspecific local adaptation versus insensitivity towards climatic conditions (Figs. 1B versus 1C), (Figs. 1C versus 1D) and the latter the varieties' capability to survive in future climate conditions. We conducted the experiment across four growing seasons using a network of botanic gardens in Finland, Norway, and Estonia (Fig. 1). Two of the experimental gardens represent the home environments of each variety, forming the reciprocal component in the experiment (Oulu and Svanvik; Fig. 1). Gardens outside of the range of the species (Tartu, Helsinki, Rauma; Fig. 1) were included to simulate the potential future climates. We tested the effect of the variety and the environmental conditions (garden or annual temperature) and their interaction on the performance of plant individuals.

Specifically, we hypothesised that

1. The varieties are adapted to local climatic conditions and thus perform better in their home environments (Fig. 1B).
2. Both varieties perform less well in the out-of-range environments, but the southern variety would have higher performance than the northern one in locations with higher temperatures outside the species' natural range (Fig. 1B).

Equally high performance of both varieties in the test environments (Fig. 1C) would reveal a lack of local adaptation, while no affinity towards conditions within the species range should result in an equally high performance of the species within and outside of its range (Fig. 1D). However, because climate change has already proceeded (*IPCC, 2019*), the process of adaptation may be lagging behind and populations may in fact be maladapted to current climatic condition (*Anderson & Wadgymar, 2020*; *Kooyers et al., 2019*; *Wilczek et al., 2014*). This scenario of climate change induced maladaptation (Fig. 1E) needs to be taken into account in contemporary translocation studies. Our study set-up thus allowed us to disentangle climatic affinity and infer the occurred and predicted effects of global warming on these two populations of the Siberian primrose.

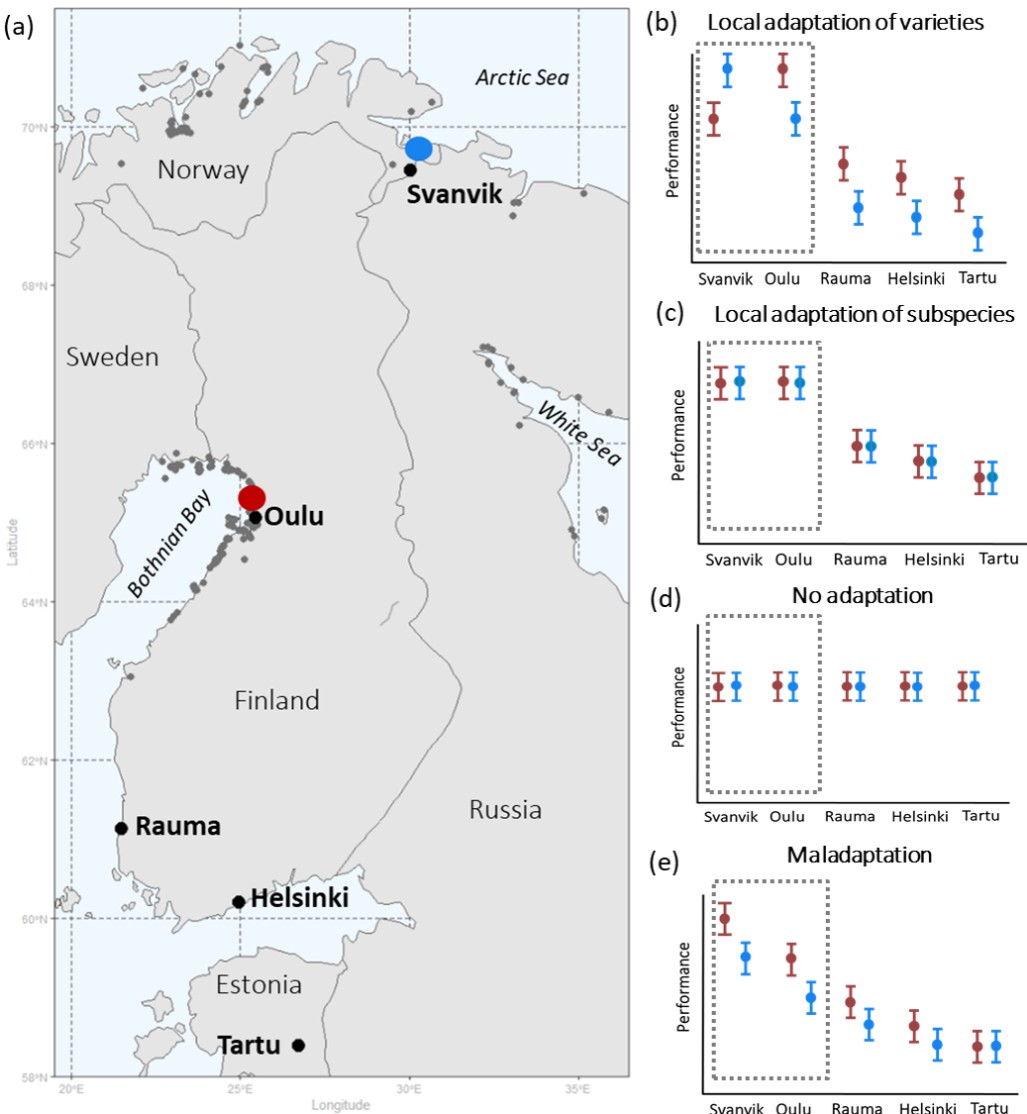

**Figure 1** **Geographical distribution of seed sampling sites and experimental gardens (A), and hypotheses of plant performance in experimental gardens (B–E).** (A) shows the geographical distribution of seed sampling sites and experimental gardens with occurrences of *Primula nutans* ssp. *finmarchica* marked by dark grey points: Var. *finmarchica* occurs by the Arctic Sea in N-Norway and var. *jokelae* by the Bothnian bay in Finland and Sweden and the White Sea in Russia. Red, seed sampling sites of the southern variety (var. *jokelae*) in Finland; Blue, seed sampling sites of the northern variety (var. *finmarchica*) in Norway. Occurrence points are available in the Supplementary Data published in Dryad (https://doi.org/10.5061/dryad.3n5tb2rfk). The source for occurrence points are: Global Biodiversity Information Facility (*GBIF, 2013*); Kastikka (Finnish plant distribution database; *Lampinen, Lahti & Heikkinen, 2012*); records from the of Finnish Environment Institute; occurrences in Russia based on information from herbarium specimens (from collections in the herbarium of the Finnish Museum of Natural History [H *sensu Thiers, 2016*] and the herbarium of the University of Turku [TUR *sensu Thiers, 2016*] and manually included occurrence points based on visually inspecting the distribution map by *Hultén & Fries (1986)*). (B–E) show hypothesized overall performance of the tested varieties in all experimental gardens following opposing underlying scenarios of (B) local relative (continued on next page...)

**Figure 1 (…continued)**
adaptation (*sensu Brady et al., 2019*) at the varietal and subspecies level, (C) relative adaptation of the subspecies to its current environment vs. areas outside it, (D) tolerance (through plasticity) towards all tested conditions (including those not currently present within the occurrence area of the subspecies), and (E) relative maladaptation caused by climate change (see text for hypotheses). Dashed area demarks within-range gardens, i.e., the reciprocal part of the experiment. Red, southern variety; blue, northern variety.

## MATERIALS AND METHODS

### Study species and seed sampling

Siberian primrose (*Primula nutans*) is a small-statured perennial herb with a discontinuous, circumpolar distribution. The Fennoscandian subspecies *P. nutans* ssp. *finmarchica* (Jacq.) Á. Löve & D. Löve is a red-listed species (VU in Norway and NT in Finland with a disjunct distribution (Fig. 1). It comprises two morpho-ecological varieties (*Mäkinen & Mäkinen, 1964*): var. *finmarchica* (the northern variety) grows on the shores of the Arctic Sea, while var. *jokelae* (the southern variety) occurs by the Bothnian Bay and the White Sea (Fig. 1A). It is a habitat specialist that mainly grows in seashore and riverside meadows (*Kreivi, Aspi & Leskinen, 2011*; *Mäkinen & Mäkinen, 1964*). The habitat preference of the Siberian primrose is believed not to be affected by specific habitat requirements - it is neither a halophyte nor does it require regular flooding (*Mäkinen & Mäkinen, 1964*). Rather, its occurrence in these habitats is likely due to its poor ability to compete in combination with a tolerance to both salty conditions and flooding. It propagates sexually by seeds and vegetatively by stolons growing from the wintering buds and the rosettes (*Mäkinen & Mäkinen, 1964*). On the basis of matrix models published by *Björnström et al. (2011)* individuals especially sterile rosettes can be relatively persistent and hence live for several years. The species is insect-pollinated and produces copious seeds that spread by gravitation, via water flows, and possibly also with birds (*Ulvinen, 1997*).

Due to the geographic distance (c. 400–550 km) between its two varieties var. *jokelae* and var. *finmarchica*, and the lack of obvious adaptations for efficient gene flow between the areas, the varieties are presumably genetically isolated, which is also reflected in their genetic differences (*Kreivi, Aspi & Leskinen, 2011*). The main areas where the varieties occur are climatically different (*Hällfors et al., 2016*; also see Fig. 2A), and there is a clear difference in the requirement of colder night conditions for flower induction in the northern variety compared to southern variety (*Mäkinen & Mäkinen, 1964*).

We sampled seeds from wild populations of both varieties of *Primula nutans* ssp. *finmarchica* from August to September 2012 (Fig. 1A, Table S1). Seeds of the southern variety were collected in five sites in Haukipudas and Ii in Finland, and seeds of the northern variety from six sites in Sør-Varanger in Norway. Seed sampling in Finland was approved by the Centre for Economic Development, Transport and the Environment in North Ostrobothnia, Finland, on July 7th, 2012 (approval number: POPELY/346/07.01/2012). For sampling seeds in Norway, no permit was needed as the species is not protected in Norway. Voucher specimens from each sampling site are deposited at the herbarium of the Finnish Museum of Natural History (H *sensu Thiers, 2016*). From each site we collected seeds from 10–53 individuals (as available, following seed collecting guidelines;

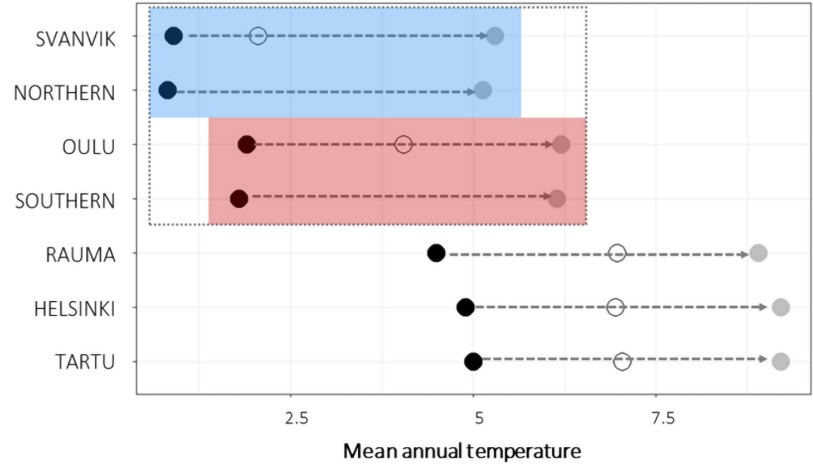

**Figure 2   Mean annual temperature of seed sampling sites and experimental gardens.** Black points show historic mean temperatures (1970–2000; 10 min resolution; (*Fick & Hijmans, 2017*) and gray points show the future projections of mean temperatures (CMIP5 for 2050, 10 min resolution, HADGEM2-ES model; (*Fick & Hijmans, 2017*)) for each experimental gardens and seed sampling sites (Northern, seed sampling sites of the northern variety; Southern, seed sampling sites of the southern variety). For experimental gardens, open circles show mean temperature during the experimental years 2013–2016. The experimental gardens and seed sampling sites within the species range are outlined by a dashed gray box, whereas the blue box indicates the sites within the range of the northern variety and the red box the sites within the range of the southern variety. Corresponding figures for precipitation sum and other climatic variables in Fig. S4. The future temperatures in Oulu approach the historic ones in Tartu, Helsinki and Rauma, and the future temperatures in Svanvik approach the historic ones in Oulu. During the experiment, mean temperatures were higher than the historic means. The conditions in Svanvik corresponded with historical means in Oulu. Those in Oulu approached the historic means in the southern gardens.

*ENSCONET, 2009*; Table S1) to obtain a representative sample. The number of seeds obtained from the collected capsules of each sampled individual varied from 0 to *c.* 200. We stored the seed lot from each maternal individual separately and left them to dry and ripen at room temperature for a minimum of two weeks after which we placed them in a freezer (−18 to −20 °C) for cold stratification to break dormancy. The seeds were kept in these conditions for about six months, until used to produce plant material for the translocation experiment (in spring 2013).

## Experimental design and plant material

We set up a common garden experiment in 2013 in five botanic gardens located in Estonia and Finland, and on research station grounds in Norway (Fig. 1A; Table S2). We chose to use botanic gardens as testing grounds instead of natural sites to avoid genetic contamination of natural populations by introducing alien genotypes, as well as for logistical reasons and legal restrictions of introducing species outside their natural range.

We chose the seeds for producing experimental plants for the trial through a hierarchical randomized method (Table S1; Fig. S1). For each experimental garden, we selected material from three seed sampling sites per variety. From each site, we randomly chose five maternal individuals. We anticipated that not all seeds would germinate and therefore we sowed

eight seeds per maternal individual (120 seeds per variety for each site) although we needed only three seedlings per maternal individual (45 seedlings per variety for each site).

We used the F1 offspring for our tests for two main reasons. Producing a 'refresher' generation between collecting the seeds and growing the experimental plants would have been difficult or impossible in uniform conditions as the two varieties flower in different temperatures conditions (*Mäkinen & Mäkinen, 1964*). Second, even in the best case, this would have delayed the experiment by at least one, possibly 2–3 years, due to the life history of the focal species. Nevertheless, a recent meta-analysis (*Yin et al., 2019*) concluded that perennial plants show hardly any transgenerational responses (i.e., effects on the offspring of the ancestor environmental conditions), whereby it is not likely that the use of F1 offspring significantly affected our results.

We sowed each seed in a 6 × 6 cm pot with commercial sowing soil (Kekkilä) mixed with sand, vermiculite and perlite (5:2:2:0.5 litres, respectively), and with a thin layer of sand on top. The seeds were sown in six cohorts, five weeks before intended planting at each experimental garden (Table S2). They were left to germinate in a greenhouse in Kumpula Botanic Garden in Helsinki, first in a small greenhouse inside the actual greenhouse to enable suitable temperatures during early spring. Temperature conditions varied between 18 and 24 °C and lighting was set to 12 h of extra light (400 W) during daytime, in addition to the day light reaching the plants through the greenhouse walls.

Due to poor germination and growth, possibly caused by supraoptimal temperature or a lack of nutrients or light, the seedlings were moved out of the small greenhouse into the actual one to allow cooler temperatures and more light (10–18 °C; Table S2). The moving followed the same order and time intervals as the sowing so that each cohort spent the same amount of time in the small greenhouse. Five days after the move, the pots were given a one-time nutrient addition with a general commercial fertilizer (Kekkilä Taimi-Superex). This is a common procedure when using nutrient poor germination soil. Without fertilization of the soil after seed germination the growth usually comes to a halt.

For each experimental garden we selected 45 seedlings of each variety, so that each variety was represented by three seed sampling sites, each of which contributed five seedlings from three different maternal individuals (Fig. S1). If there were not enough seedlings from a certain maternal individual, we complemented the design with seedlings from another maternal individual of the same seed sampling population (in 69 cases). If there was no other such maternal individual, we used seedlings from another maternal individual from another seed sampling population but of the same variety (in 21 cases).

Because of a delayed spring in 2013, establishing the experimental plots had to be postponed by three weeks to avoid frozen soil and frost that could kill the seedlings after planting. The approximately two-month-old seedlings were transported and planted during May–June 2013 (see Table S2 for exact dates), starting from the southernmost garden in mid-May (Tartu, Estonia) and reaching north (Svanvik, Norway; Table S2) in late June. The intention was to follow the advance of the season so as to plant as early as possible while avoiding frost damage. In each experimental garden, we planted the seedlings in three 165 × 145 × 20 cm experimental plots with a minimum distance of 30 m between the plots, following a randomized block design. A filter cloth was placed at the bottom to

prevent the ground soil from affecting the growing conditions and to help retain moisture. The plots were filled with fine sand and peat mixture (75 and 25 volume-%, respectively) with some added dolomite lime. The intention was to roughly mimic the seashore meadow soils where the species grows in the wild while using commercially available soils that could be obtained in large quantities by a commercial soil provider and transferred to the all experimental gardens. The aim with using the same substrate was to homogenise (to certain extent) the growth conditions and thus to be able to separate the thermal effect from other effects.

In each plot, we planted 30 seedlings (i.e., 90 seedlings per garden) in a grid, c. 21 cm from each other and 30 cm from the wooden frame. In each plot, 15 seedlings were of the southern and 15 of the northern variety, placed in a random order. We placed light metal cages on top of each plot to prevent large animals from interfering with the experiment. Plots were irrigated with *c.* 20 litres of water once a week during the growing season to keep the soil sufficiently moist. Although precipitation is part of climate, the effects of which we attempt to measure, we considered basic watering of the trials necessary as this is a sea-shore plant naturally exposed to high water tables. This watering regime was not intended to directly resemble the seashore meadows in nature, where the water level can vary substantially and occasionally even flood the plants. Instead, we applied irrigation to avoid large effects of drought, which this seashore species growing on occasionally inundated meadows, would not endure. Thus, watering was done only to the amount to compensate for the fact that our test sites were not at the shoreline, i.e., to mimic the natural sites of the primrose up to a certain baseline so that the plants would not die because of an unnatural unfavourable moisture regime. Plots were also weeded when deemed necessary to avoid effects of competition from other species, as our focus was not on the effect of competition on plant performance.

Since almost 50% of the plants died during the first summer (139 individuals of the northern variety and 57 of the southern; likely due to small seedling size and too little irrigation), in the autumn of 2013, the experimental plots were supplemented with left-over plants that had been growing outdoors in Kumpula botanic garden in Helsinki during the summer (dates presented in Table S2). We chose new plants repeating the plant selection process described above for situation where there were not enough representatives of the preselected maternal individual for the experimental garden. We recorded the planting time, to enable differentiating between original and new plants in subsequent analyses. Altogether, 614 individuals were planted in the five gardens during the early and late summer visits.

To describe the performance of individuals, we measured their survival, size, and flowering (whether they flowered, how many flowers they produced, and when) from year 2014 to 2016. The fitness of genotypes can be described as the relative success with which they transmit their genes to the next generation (*Silvertown & Charlesworth, 2001*). Because of the limited temporal extent of our study and because we do not measure the individual's fitness through its ability to produce viable progeny, we focus on key plant performance measures likely to correlate with fitness. Survival is a definite measure of fitness as a dead individual cannot transmit genes, but survival can also be stochastic. Also, the physical size

of an individual and reproductive output, such as abundance of flowering, tend to correlate with plant fitness (*Silvertown & Charlesworth, 2001*). Each spring (2014, 2015, and 2016) we recorded flowering presence and abundance. Flowering was inventoried approximately every second day for 14 days after the first flower appeared in that garden in the specific year. In the autumn of the same years, we recorded survival and photographed each surviving plant from above. We assessed plant size from photographs, through digitally cutting out and measuring the area (cm²) of the visible vegetative parts. We used the size at the time of planting as a measurement of original size, but 24 individuals lack data on that due to missing photographs or insufficient resolution. These individuals were therefore not included in the analyses, and our total N is reduced from 614 to 590.

## Weather and climatic data

For describing the climatic conditions at the seed sampling sites and experimental gardens, we obtained data on historic climatic conditions (1970–2000; 10 min resolution) and future projections (CMIP5 for 2050, 10 min resolution, HADGEM2-ES model) through the Worldclim database (*Fick & Hijmans, 2017*). These climatic data were downloaded in R using the *getData* function in the *raster* package (*Hijmans, 2019*). Additionally, we modelled plant performance as a function of mean annual temperature in the experimental gardens during 2013–2016 (see below; Fig. 2). The weather data were obtained from the Gridded Agro-Meteorological Data in Europe (*Joint Research Centre, 2014*), which contains meteorological parameters from European weather stations interpolated on a 25 × 25 km grid. We used the function *biovar* in the package *dismo* (*Hijmans et al., 2017*) to calculate bioclimatic variables of the weather data in R.

We present historic and predicted mean annual temperature for the seed sampling sites and experimental gardens and as a mean across experimental years for each garden in Fig. 2. Mean annual temperature is also shown explicitly for each year in each site in Figs. S4 and S5 together with several additional bioclimatic variables, including annual precipitation sum and mean temperature during the warmest quarter of the year. The current climatic conditions in Tartu, Helsinki, and Rauma roughly correspond to the anticipated future climatic conditions in Oulu and in the distribution area of the southern variety (Fig. 2, Fig. S4). The historic conditions in Oulu roughly correspond to the future conditions in Svanvik and in the distribution area of the northern variety (Fig. 2, Fig. S4).

In addition to modelling plant performance as a function of experimental garden and other variables, we conducted an alternative model using weather data instead of experimental garden. To represent weather, we chose annual mean temperature at the experimental gardens across experimental years. Climate parameters tend to be multicollinear, and we do not know which specific aspects of temperature these plants respond to. Thus, mean annual temperature is used as a large-scale descriptor of differences between experimental gardens.

## Statistical analyses

To test the performance of the two varieties in the different gardens we analysed their combined performance using aster models (*Geyer, Wagenius & Shaw, 2007*; *Shaw et al.,*
*2008*). These models allow the joint analysis of several life history components that together quantify differences in performance, such as survival, flowering and flowering abundance. The complete model (Fig. S2) included plant survival and flowering in each year (0 or 1; Bernoulli error distribution) plus flowering abundance (zero-truncated Poisson distribution of error) that were conditional on survival in the previous year. Whether a plant flowered or not was also conditional on it having survived in the year of flowering. The abundance of flowers per individual was further conditional on that the individual flowered that year.

These combined responses were modelled using the *aster* call in the aster package (*Geyer, Wagenius & Shaw, 2007*; *Geyer, 2017*) in two main models: one testing the effect of *Garden,* and one testing the effect of *Temperature* (mean annual temperature over 2013–2016). Specifically, the responses were modelled as a function of *original size* at planting (continuous, square-root transformed and thereafter centred around the mean of each variety; see Fig. S3 for distribution of original size per variety and experimental garden), *Variety* (categorical: Southern and Northern) and either *Garden* (categorical, 5 levels) or *Temperature* (continuous) and the interaction between *Variety* and either *Garden* or *Temperature*. To rule out the effect of *planting time* (early or late summer) we also ran a model including this variable, which did not improve model fit (change in deviance 1.96, change in $df = 1$, $p = 0.16$). The estimated mean value parameters represent overall flowering output, that is, the expected flower count during the whole experiment for plants of average original size (*Geyer, 2019*). We estimate these for each variety at each experimental garden and hereafter refer to these predicted values as the overall performance.

As we assumed random variation between the three *Plots* in each garden and between the five to six *Seed sampling sites* of each variety, we also fitted random-effect aster models for both main fixed effects (i.e., for the *Garden* and *Temperature* model) using the *reaster* function in the *aster* package (*Geyer, 2017*). Predicting plant performance based on random effects models is not straightforward nor directly possible within the aster models package (*Geyer, 2019*; *Geyer Charles et al., 2013*; although see *Geyer Charles et al., 2010*). Nevertheless, as the estimates for the fixed effects were very similar for both the aster models including only fixed effects and the models also including random effects (see Table S4 for the Garden-models and Table S5 for the Temperature-models) we obtained predicted values of plant performance based on the fixed effects models using the *predict* function in the *aster* package (Tables S6 and S7). Finally, for assessing statistical significance of fixed effects, we conducted likelihood ratio tests (anova) of the full random effects model versus models excluding individual variables (Table 1). To interpret the role of mean annual temperature for explaining differences in performance between the varieties, we additionally calculated deviances in mean annual temperature for each site in contrast to the historical mean annual temperature in the home site for both varieties.

## Summarizing results using local adaptation metrics

For the reciprocal part of the trial, we calculated local adaptation metrics from the mean estimated combined predicted values obtained through the fixed effects Garden-model

**Table 1  Change in deviance when a single variable was omitted from the full random effects models.** Summary from comparing aster models testing the fixed effects of experimental garden (lefthand side of table) or annual temperature (right hand side of table), variety and original size on plant performances, as well as *Plot* and *Seed sampling* site as random effects. Statistical significance of predictor variables was assessed using likelihood ratio tests, that is, a model where one variable was omitted was compared to the full model using anova.

| Garden-model | | | | Annual temperature -model | | |
|---|---|---|---|---|---|---|
| *p*-value | df | Change in deviance | Variable | Change in deviance | df | *p*-value |
| <0.001 | 1 | 62.4 | Original size | 60.2 | 1 | <0.001 |
| <0.05 | 4 | 11 | Variety:Garden/Ann.Temp. | 8.8 | 1 | <0.01 |
| <0.01 | 5 | 20 | Variety (+ Variety:Garden/Ann. Temp.) | 17.2 | 2 | <0.001 |
| <0.05 | 8 | 19.6 | Garden/Ann. Temp. (+ Variety:Garden/Ann. Temp.) | 14.2 | 2 | <0.001 |
| <0.001 | 0 | 46 | Plot | 60 | 0 | <0.001 |
| 0.09 | 0 | 1.8 | Seed sampling site | 2.2 | 0 | 0.07 |

(see the mathematical formulas below and Table S6). We did not include the out-of-range experimental sites in order to specifically enable quantification of local adaptation patterns of the two varieties within the distribution area of the variety. This gives an indication of the current situation of local adaptation, while including the out-of-range gardens would confuse the interpretability of the metrics over time and space. There are three main approaches to make quantitative comparisons of local adaptation from performance measures (*Blanquart et al., 2013*). The sympatric-allopatric contrast (henceforth called ΔSA) focuses on the *difference* between the average performance of the genotypes in their home environment and non-home environment. Thus, it describes how well the genotypes overall fit their home-environments. The home vs. away approach (*HA*) in turn describes the *home environment quality* for each tested genotype, and is measured by the difference between the fitness of a genotype in its home environment and the away environment. Finally, the local vs. foreign approach (*LF*) describes the *genotype quality* for each tested home environment, and is measured by the difference between the fitness of a genotype in its home environment and the mean fitness of all other genotypes when exposed to the same environment. These three conceptual approaches relate to different aspects of local adaptation, all of which lie in the interest of our study: the overall habitat quality, the overall quality of the variety, and how the two varieties fit the different environmental conditions. Below, we provide a brief description of the metrics that we used, while a more thorough definition can be found in Text S1 and in *Blanquart et al. (2013)*.

We estimated the mean performance for an average sized plant of each variety in each experimental garden over four years (based on the aster model predictions). We used these estimates to calculate local adaptation metrics. Specifically, ΔSA for the *subspecies as whole* is the difference between the mean performance (*p*) of both varieties (northern, N, southern S) in their home (H) environments and that in the test (away, A) environment:

$$\Delta SA = \frac{p_{NH} + p_{SH}}{2} - \frac{p_{NA} + p_{SA}}{2}$$

The Home vs. Away metric (*HA*) for each *variety* is the difference between the performance

of each variety in its home environment and that in its away environment.

$$HA_S = p_{SH} - p_{SA}$$
$$HA_N = p_{NH} - p_{NA}$$

The Local vs. Foreign metric ($LF$) for each *garden* is the difference between the performance of the local variety and that of the foreign variety.

$$LF_S = p_{SH} - p_{NA}$$
$$LF_N = p_{NH} - p_{SA}$$

The performance estimates range between 0.04 and 5.7 and were therefore scaled to range between 0 and 1. The $HA$, $LF$ and $SA$ metrics will, therefore, range between -1 and 1, with values around zero indicating negligible differences in performance. The closer the value is to 1, the stronger is the evidence for positive response of the local genotype in the environment, or suitability of the environment for the genotype, and values close to -1 show the opposite.

All data management and analyses were conducted in R (version 3.5.3; *R Core Team, 2019*).

## RESULTS

### Realized weather conditions during the experimental years

The weather conditions during the experimental years differed between the experimental gardens in accordance with what we expected when designing the experiment (Fig. 2, Fig. S4). However, the observed temperatures were warmer than the historic means. In essence, this means that the experienced temperature conditions in Svanvik resembled the historic average temperatures in Oulu (deviation of 0.146 °C), while the experienced temperatures in Oulu resembled, or were even warmer than, the historic average temperatures in the out-of-range gardens in Rauma, Helsinki, and Tartu (Fig. 2). The northern variety, however, did not experience conditions corresponding to historical temperature in its home site in any of the experimental sites, as even the coldest site, Svanvik, deviated from its historic mean annual temperature by +1.146 °C. The variation in bioclimatic variables between gardens and years are shown in Fig. S5. Additional variables describing the weather conditions during the historic period and experimental years are provided in Figs. S4 and S5.

### Plant survival

Altogether, 110 out of 590 (18.6%) plant individuals survived in the experiment by 2016. Low survival is common in transplantation experiments (*Dalrymple et al., 2012*). However, there were large differences in survival between the varieties and the gardens in our experiment. The survival across gardens was 8.8% (27/307) for the northern variety and

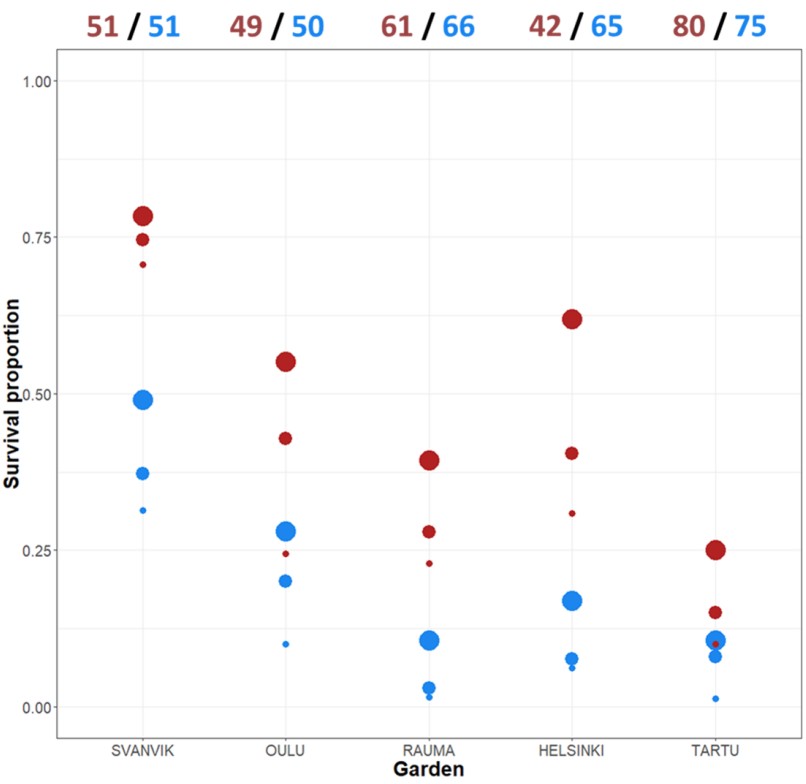

**Figure 3  Mean survival per variety across gardens and years.** The total number of individuals planted in 2013 appear at the top of the plot (both original plants and additional plants due to high die-off in the first summer). Red points, southern variety; blue points, northern variety. The size of the point indicates year: big point, 2014; medium point, 2015; small point, 2016. Please refer to Table S3 for the exact numbers of individuals per year and garden, and percentage decrease over year.

29.3% (83/283) for the southern variety. The survival varied between the gardens from 5.8% in Tartu (9/155) to 51% in Svanvik (52/102; Fig. 3, Table S3).

Individuals of the southern variety survived at a higher percentage every year and in each garden (Fig. 3; Table S3), yet, the difference in survival between the varieties varied according to the garden. Both varieties showed roughly the same trend, surviving worst in the out-of-range gardens (Rauma, Helsinki, and Tartu) and best in the northernmost garden (Svanvik) (Fig. 3; Table S3).

## Flowering and plant size

Mean values of our data on flower presence, flowering abundance and size (cm²) are presented in Fig. S6, both across all planted individuals and across all individuals that were alive in the focal year. Among all the planted individuals, flowering was most frequent and abundant, and plants grew biggest in Svanvik. The southern variety also grew well and flowered abundantly in Helsinki. When considering only the surviving individuals, however, both varieties flowered frequently and abundantly in Tartu and Rauma and the surviving individuals of the southern variety grew big in all southern gardens. Yet, in many

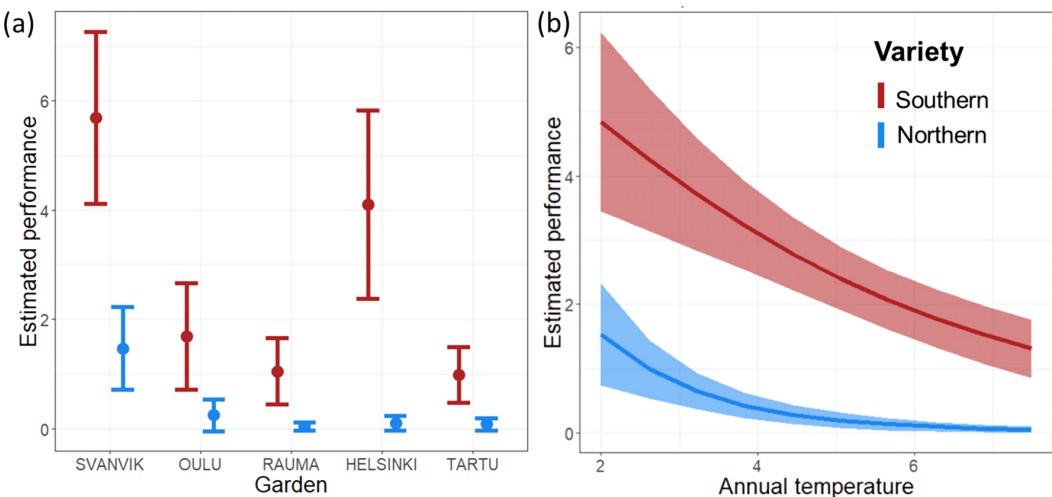

**Figure 4** **Predicted overall performance of the two varieties across (A) experimental gardens, and (B) mean annual temperature over the experimental period 2013–2016.** Error bars represent 95% confidence intervals. Predictions are made for plants of average original size and based on the fixed effects aster model (i.e., not including *Plot* and *Seed sampling site* as random effects). Red, southern variety; blue, northern variety.

cases the number of surviving individuals was very low, for example, only one individual of the northern variety was alive in Tartu in 2016 (Table S3), but it flowered.

## Overall performance

Both the Garden and Temperature models revealed similar and significant differences between the varieties in their overall performance (Table 1; Fig. 4). The Garden model indicated increased performance of both varieties in gardens towards the north but revealed substantial differences between the varieties (Fig. 4A; Table S6). Likewise, the Temperature model revealed a significant negative effect of mean annual temperature on the performance on both varieties, with a stronger negative effect on the northern variety (Fig. 4B; Table S7). Indeed, we found significant interaction effects between variety vs. experimental garden (Fig. 4A; Table 1) and annual temperature (Fig. 4B; Table 1). In addition, the original size at planting had a highly significant effect on plant performance in both models (Table 1). The varieties are morphologically different, including in size: the southern variety is generally bigger than the northern variety (*Mäkinen & Mäkinen, 1964*). As described in the methods, this was taken into account by centering the variables describing original size around the mean of each variety, wherefore the response in overall performance as a function of original size measures the relative effect of size per variety.

Of the random effects, *Plot* explained a significant proportion of the residual variance (change in deviance was 40 and 60 when Plot was omitted from the model for the Garden and Temperature models, respectively) while *Seed sampling site* explained a marginal and non-significant proportion of the residual variation (Table 1; Tables S4 and S5).

The aster models provide an overall performance estimate based on flower abundance per variety in each garden, conditional on survival and presence of flowers. For the southern

variety, these estimates (Fig. 4, Tables S6 and S7) were highest in Svanvik, followed by Helsinki (however with overlapping standard deviations of estimates; Fig. 4 and Table 1). The overall performance of the northern variety corroborated our hypothesis (Fig. 1B) by showing the highest performance in its home-environment Svanvik, followed by Oulu, and performing worse in all three out-of-range gardens (Tartu, Helsinki, and Rauma). An additional model on the deviances of temperature from the historical mean instead indicated that the less deviance, the better both varieties perform (Fig. S7). However, the southern variety still out-performed the northern variety, indicating that other factors in addition to climate are important for explaining the performance of these varieties.

### Local adaptation metrics

The local adaptation metrics based on the reciprocal part of the experiment (Tables S6 and S8) suggested maladaptation to the current home environments of both varieties. For the southern variety, the *HA*-value of -0.71 indicates that it experiences reduced habitat quality in its home environment, as it thrives better away (in Svanvik) than at home (in Oulu). The *LF* of 0.26, however, indicates that the southern variety possesses superior quality in Oulu since it still does better at home than the tested foreign northern variety does in Oulu. For the northern variety, the *HA* of 0.22 indicates that it experiences superior habitat quality in its home environment (Svanvik) compared to the tested away-environments (Oulu). Nevertheless, the *LF* of $-0.75$ indicates that this variety is of reduced quality in its home environment, as the foreign (southern) variety performs better there. Mean $\Delta SA$ for the subspecies as a whole was -0.25 indicating an overall negative fit of the environments for the subspecies.

## DISCUSSION

Our results reveal clear patterns of adaptation to macroclimatic conditions at the subspecies level. In general, both varieties showed their best performance within the species' current range. The southern variety showed higher overall performance in all experimental gardens, compared to the northern variety. The overall survival and size as well as flowering frequency and abundance tended to be higher for the southern variety. This is largely in line with the results from a greenhouse experiment conducted by *Mäkinen & Mäkinen (1964)*. In that study, the northern variety flowered only in cold night conditions (12 h 21 °C + 12 h 5 °C versus 24 h 21 °C), whereas the southern variety flowered only in continuously warm conditions but showed increased vegetative growth in the cold night conditions. Similarly in our study, the northern variety flowered most frequently in cooler conditions towards the north and rarely in the south.

We did not find characteristic indications of local adaptation at the variety level, when looking at the performance of the sampled populations per experimental garden. Individuals of both varieties survived best, and flowered more frequently and abundantly, in the northernmost garden Svanvik. This pattern is also evident from the local adaptation metrics, where the *LF* metric indicated that the southern variety was superior in the home environment of the northern one. Further, the overall performance proxy, the sympatric-allopatric contrast, indicated that the varieties may have become maladapted
to the conditions that prevail at present in their current environments. This suggests that the conditions have already changed, compared to historical conditions to which the populations were better adapted. In other words, the varieties are suffering from a so-called adaptational lag (*McGraw et al., 2015*). When considering the weather conditions during the experimental years, compared to the historic climatic conditions (Fig. 2B, Fig. S4), a conclusion of local adaptation to historical temperature conditions indeed becomes the most parsimonious explanation. This is revealed by the fact that the historical mean annual temperature of the southern variety was most closely realized in Svanvik during the experimental years, while the historical mean annual temperature for the northern variety was not realized in any of the experimental gardens. In addition, the mean annual temperature in Oulu resembled that of the three southern gardens' historical means, which was the climate change treatment that we had envisioned. Thus, the conditions that we aimed to mimic through our experimental design had in practise shifted locations one step further north, which we did not take into account initially, when verbalising our hypotheses. Our findings are also supported by a species distribution modelling study that tested the difference between modelling at the level of variety and at the level of subspecies (*Hällfors et al., 2016*). According to the conclusions of that study, the southern population, if indeed locally adapted to the historical conditions in its current range, should already by the 2030s find suitable conditions in the current distribution of the northern variety, while the same conditions will no longer remain suitable for the northern variety.

Maladaptation is presumably increasing due to climate change, as the rapid change outpaces the ability of species to adapt or migrate, and conditions reach beyond the extent of temperature tolerance (*Crespi, 2000*; *Kooyers et al., 2019*). Our findings are in line with studies showing signs of maladaptation to current climatic conditions in plant species (see *Anderson & Wadgymar, 2020*; *Gellie et al., 2016*; *Kooyers et al., 2019*; *Wang et al., 2019*; *Wilczek et al., 2014*). *McGraw et al. (2015)* found that the climatic optimum of their study species had shifted by 140 km. Our study reveals a displacement over a much longer extent: the distance between the seed sampling sites of the southern variety and the Svanvik experimental sites is over 500 km.

Although maladaptation may be increasing because of climate change, it can also be caused by underlying population genetic structures, such as genetic drift or founder effects (*Crespi, 2000*; *Leimu & Fischer, 2008*) related to the biogeographic history of the populations. The two varieties of Siberian primrose probably became spatially separated during the end of the last glacial period (*Mäkinen & Mäkinen, 1964*), which first allowed colonization of emerging areas and later fragmented the population through diminishing suitable habitat, as land masses re-formed from underneath the receding ice shelves (see *Kreivi, Aspi & Leskinen, 2011*). Indeed, genetic and allelic diversity is known to be remarkably low in all populations of the Siberian primrose in Fennoscandia (*Kreivi, Aspi & Leskinen, 2011*). However, the consistent pattern of increased performance of both varieties from south to north suggests that maladaptation caused by recent change in climate is a likely explanation for the results of this study. Maladaptation due to drift or other stochastic processes would be geographically random, and a pattern consistent with a spatial gradient would likely not appear. In addition, at least for the southern variety,
this is not only a question of relative maladaptation (for definition of terms, see *Brady et al., 2019*). According to threat assessments, it also shows absolute maladaptation in the form of population decline (*Hyvärinen et al., 2019*), results for this species available at https://punainenkirja.laji.fi/en/results/MX.38659?checklist=MR.424).

## Potential confounding effects and general limitations of transplantation experiments

While our results suggest a strong impact of climate on the Siberian primrose, they are not conclusive. Large-scale translocation experiments are complex, and it is important to evaluate potential methodological pitfalls. Transplantation experiments along the north-south axis necessarily contain confounding effects, such as day length and other non-thermal environmental conditions (*Saikkonen et al., 2012*; *Björkman et al., 2016*), which cannot be removed in a transplantation study. Despite the rigorous measures to standardise treatments in all the gardens , we had only one experimental garden at each latitude (albeit with three plots in each garden). Therefore, random variation in care, pressure from weeds, and weather events may have confounded the climatic signals. For example, some plots may have experienced short periods of drought despite our regime of additional watering designed to mimic the natural conditions experienced by this species of damp seashore meadows. The roughly three-year observation period of our study may also not fully represent the prevailing climates in the gardens, since inter-annual variation in weather may mask mean climatic differences, nor does such a short experiment necessarily reveal local adaptation (*Bennington et al., 2012*). The fact that the site-specific deviance in mean annual temperature did not explain the relative poor performance of the northern variety, indicates that other factors, like more nuanced differences in climate, photoperiodic cues, or site-specific variation had a strong effect on performance. Long-term studies would even out annual variations and allow the measurement of characteristics that become available later in a plant's life cycle and population development (seed production and germination, adaptation across generations, etc.).

In our study, plants that were larger at the time of planting showed higher overall performance. Previous studies of reintroductions have shown that the best results (the highest numbers of surviving plants) are gained by using seeds or mature plants (*Dalrymple et al., 2012*). However, in an experimental study, we believe that using mature plants can be confounding since they may be affected by the climatic conditions of the garden where they are propagated for the experiment, thus possibly biasing the results through stronger priming effects (see below for a discussion of the potential effect of Helsinki on the southern variety). Also, early stages in the life cycle may be the ones most susceptible to climate change as they define the environments to which post-germination life-stages of the individuals are exposed (*Donohue et al., 2010*), and therefore they should not be omitted from a test setting. On the other hand, while running experiments from seeds would have allowed us to evaluate performance through germination, this might have risked the experiment in case of failure to germinate. In fact, even in the relatively stable greenhouse conditions, the germination percentage of these varieties was relatively low (27–69%; depending on the sampling site; *Lehtimäki, 2016*).

The seeds of each variety were sampled in a single region within a c. 10 km radius (Fig. 1, Table S1). Thus, from the data at hand, we cannot infer patterns across the complete ranges of the varieties. Nevertheless, the genetic variation within each variety is relatively low (*Kreivi, Aspi & Leskinen, 2011*) and the varieties occur in distinct climatic environments (*Hällfors et al., 2016*), which makes it plausible to assume that individuals across the range of each variety would respond similarly to environmental conditions.

We did not control for, or try to remove, maternal carry-over effects, for instance through growing a first generation in identical and controlled conditions and then utilizing the progeny for the trials. However, the random effect of seed sampling site did not significantly increase the explanatory power of residual variation. On the other hand, the lack of replication across the distribution of the varieties inevitably lessens the potential for drawing strong conclusions about the strength of local adaptation and potential effects of climate change for the subspecies as a whole.

Another useful performance measure would have been seed production of the tested individuals from the experimental locations and subsequent germination of the seeds. However, because of limited resources and the possibility of hybridization between the two varieties, we restricted our observations to the most readily observable performance measures of the planted populations. In natural conditions each seed capsule may contain tens to hundreds of seed. Assuming that pollination was successful (the species is pollinated by a variety of insects including small flies which are present at all sites), flower number would be a relevant proxy for seed set. Although germinability of seeds is a more conclusive indicator of reproductive capacity and population fitness, flowering is also an important indicator of the existence of a necessary life cycle stage towards offspring. In addition, for this species, flowering has been shown to be an important fitness-reflecting factor, as failure in flowering is connected to increased extinction risk of the species (*Björnström et al., 2011*).

Why the southern variety thrived relatively well in Helsinki remains unclear. The raw proportion of surviving individuals in Helsinki (31%) was not substantially higher than in Oulu and Rauma (24.5 and 23%, respectively). However, the overall performance estimate, which describes population-level flowering output, was substantially higher in Helsinki (4.1) than in the variety's home environment in Oulu (1.69), and in Rauma (1.01). We suspect that this could have been caused by the above-mentioned difference in general maintenance or weather conditions that we were unable to capture through long-term means. Because the individuals were propagated together in common conditions in Helsinki, the difference between these conditions and those of the experimental garden, where the individuals ended up, could additionally explain the high performance in Helsinki. If the individuals planted in Helsinki were primed to the Helsinki conditions, this might explain the higher performance of them compared to the other southern gardens and to Oulu. However, as this higher performance was not seen for the northern variety (although the varieties may be differentially affected by priming) we suspect that the main cause was stochastic effects.

### The effect of climate change on the Siberian primrose

In this study, we have elucidated the role that plasticity towards varying temperature conditions can play within the boundaries of local adaptation for the Siberian primrose and its two varieties. We conclude that, especially for the northern variety, the existing plasticity towards warmer conditions is very limited. As the climate continues to change, migration northwards could enable the varieties of this species to cope with increasing temperatures. However, because of its poor dispersal ability, habitat specialisation further reducing chances of range expansion, and limited temperature tolerance, the available options for this species are restricted to the production of characteristics suitable for the new climatic conditions in situ through evolutionary adaptation. Nevertheless, it remains unclear how big a role evolutionary adaptation will play under climatic change in relation to temperature tolerance (*Arnold, Nicotra & Kruuk, 2019*; *Charmantier & Gienapp, 2014*; *Gienapp et al., 2008*; *Nicotra et al., 2010*). It is also not clear whether selection can act to increase or decrease plasticity to allow larger variation among phenotypes, which could then respond adaptively to a range of temperature conditions (*Arnold, Nicotra & Kruuk, 2019*).

Our raw data indicate that those individuals that did survive in the southernmost gardens tended to be big and to flower frequently and abundantly (Fig. S6). Thus, the elimination of individuals that are maladapted to the conditions in each experimental garden leaves a survivor population for which the impact of climate is not as decisive. The adaptive performance of the remnant plants in the out-of-range gardens highlights how individual features may be key determinants for continued survival when a population goes through a bottleneck (*Carson, 1990*). Whether such a small founding population of a genetically impoverished species could form a viable population under rapidly changing conditions is, however, uncertain. Experiments on the evolutionary potential in this species are needed to draw more solid conclusions on the potential for the species to overcome the lag in adaptation that it is likely currently experiencing.

## CONCLUSIONS

Siberian primrose, a habitat specialist and poor disperser with small and fragmented populations that harbour low genetic variability and a limited thermal tolerance, is an example of a species likely to suffer from the lack of adaptive capacity under rapid climatic change. Proactive conservation methods, such as assisted migration, may be needed in order to save this species. Overall, our results cause concern about the viability of the Siberian primrose, especially the northern variety, for which the detrimental effects of climate change may become evident within a few decades. In addition, these findings remind us of the need to take the population level into account when modelling the effect of climate change on species (*Souther & McGraw, 2011*; *Hällfors et al., 2016*).

The results of this study also highlight a critical point in testing for local adaptation and plasticity under the rapidly changing climate. If populations are not able to track ongoing changes, only experiments both within and outside the current range of the species (or variety) can provide accurate information regarding local adaptation and

plasticity towards historical, current, and future conditions. Local adaptation to historical conditions, which currently may prevail only polewards and even outside of the species range, must be considered a possible scenario (e.g., *Anderson & Wadgymar, 2020*; *Gellie et al., 2016*; *Kooyers et al., 2019*; *McGraw et al., 2015*; *Wilczek et al., 2014*), in which case the home environment no longer represents the "home climate". The out-of range gardens that were included in this study (Rauma, Helsinki, and Tartu) can inform us about likely trajectories for future maladaptation in case the species is not able to adapt evolutionarily and climate continues to shift temperature isoclines. Climate change thus gives cause for re-examining traditional reciprocal and out-of-range common garden experiments (*Anderson & Wadgymar, 2020*), and to plan experiments with climatic and weather data at hand. Comparing responses to future, current, and historical conditions offers a powerful approach for future studies (*Anderson & Wadgymar, 2020*; *Franks, Sim & Weis, 2007*; *Merilä & Hendry, 2014*; *McGraw et al., 2015*; *Thomann et al., 2015*). Together with continued monitoring of population abundances (*Cotto et al., 2017*), such studies can provide early warning signals for potential feed-back effects on the ecosystem productivity (*Schedlbauer et al., 2018*; *Curasi et al., 2019*) and species that may be close to tipping points and to going extinct.

## ACKNOWLEDGEMENTS

We are grateful to the botanic gardens that provided staff and premises for conducting the trials: University of Tartu, University of Helsinki, University of Turku teaching garden in Rauma, University of Oulu, and NIBIO Svanhovd. We thank Pertti Pehkonen, Outi Pakkanen, and Toomas Kangro for technical assistance in preparing the experiments. Terhi Ryttäri and Henry Väre gave advice on the species and required soil properties. Ritva Hiltunen and Tuomas Kauppila assisted in collection of seeds, Elina Vaara helped with permits for seed and plant transfers, Hanna Finne with data management, and Bess Hardwick with acquiring weather data.

### Funding

Maria Hällfors was supported by the University of Helsinki Research Fund, LUOVA—Doctoral Programme in Wildlife Biology Research and the Jane and Aatos Erkko Foundation through the Research Centre for Ecological Change, University of Helsinki. The study was supported by the Academy of Finland grant 126915 and Societas pro Fauna et Flora Fennica. The funders had no role in study design, data collection and analysis, decision to publish, or preparation of the manuscript.

### Grant Disclosures

The following grant information was disclosed by the authors:
University of Helsinki Research Fund.
LUOVA—Doctoral Programme in Wildlife Biology Research.

Jane and Aatos Erkko Foundation.
Academy of Finland: 126915.
Societas pro Fauna et Flora Fennica.

## Competing Interests

The authors declare there are no competing interests.

## Author Contributions

- Maria Hällfors conceived and designed the experiments, performed the experiments, analyzed the data, prepared figures and/or tables, authored or reviewed drafts of the paper, and approved the final draft.
- Susanna Lehvävirta, Leif E. Schulman and Marko T. Hyvärinen conceived and designed the experiments, performed the experiments, authored or reviewed drafts of the paper, and approved the final draft.
- Tone Aandahl, Iida-Maria Lehtimäki, Lars Ola Nilsson and Anna Ruotsalainen performed the experiments, authored or reviewed drafts of the paper, and approved the final draft.

## Field Study Permissions

The following information was supplied relating to field study approvals (i.e., approving body and any reference numbers):

Seed sampling in Finland was approved by the Centre for Economic Development, Transport and the Environment in North Ostrobothnia, Finland, on July 7th, 2012 (Diary number POPELY/346/07.01/2012).

For sampling seeds in Norway, no permit was needed as the species is not protected in Norway.

## Data Availability

The data are available on Dryad: Hällfors, Maria et al. (2020), Data for: Translocation of an arctic seashore plant reveals signs of maladaptation to altered climatic conditions, Dryad, Dataset, https://doi.org/10.5061/dryad.3n5tb2rfk.

## Supplemental Information

Supplemental information for this article can be found online at http://dx.doi.org/10.7717/peerj.10357#supplemental-information.

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
