# Peer review of "Translocation of an arctic seashore plant reveals signs of maladaptation to altered climatic conditions"

_PeerJ, doi:10.7717/peerj.10357_

## Round 0.1 · original submission · Major Revisions

· Academic Editor

Major Revisions

Three independent reviews have been received, which all are positive to the manuscript. However, some critical concerns are raised by the reviewers that need to be considered carefully. Particularly Reviewer 1's comments on the experimental design and the findings of the maladaptation have to be addressed and the main text needs to include more of the experimental design and caveats rather than having this information only in the supplemental methods. Furthermore, I also think the authors should consider at least some of the suggested references by Reviewer 3, but also his comment on the use of numbers of flowers as a measure of fitness need carefully consideration.

Reviewer 1 ·

Basic reporting

This manuscript investigates the whether varieties of a perennial herb are locally adapted or exhibit signatures of maladapted due to changing climates. The experiment(s) presented here are designed well, the paper is well written, and the results are interesting, important and generally on point. The number of years that the experimental gardens were maintained was impressive and provide confidence in their conclusions both for the basic science and management aspects of the project. This is the clearest signal of adaptation lag that I have seen in the literature. However, I do have a number of concerns and comments that hopefully will clarify the experiment design, results and associated conclusions for the reader.

Experimental design

Reading through the supplemental methods describing the design of the experiment, I grew a bit concerned about whether this experiment actually measures local adaptation. The plants were given excess fertilizer, had limited contact with the native soil at each site, and were given water on a weekly basis to prevent die off from drought. It seems like what was really measured was how well each variety performed in the different temperature and herbivore regimes of each site. Even so, temperature may have a larger effect than reported as higher temperatures could facilitate more substantial droughts. I fully realize that all field common garden studies have these issues to a certain degree, but I suggest that more of the experimental design be reported in the main text and the issues I raise should be a part of the discussion.

I am a bit confused about why dead plants were replaced at the end of the 2013 growing season with mature plants and how these plants were taken into consideration in the analyses. Please provide more explanation (can be in the supplement if they were not included in any of the analyses). However, I think these were somehow included in the analysis (i.e. Line 260) despite that they did not go through the most severe bottleneck in the experiment (i.e. the first year).

As discussed in the methods, results and Figure S4, southern variety plants were larger at planting than northern variety plants. This size is included in the aster models as a highly significant covariate. However, I am not sure whether this covariate’s contribution to fitness is fully accounted for in the likelihood ratio significance tests within the aster model. In a standard ANOVA hypothesis testing framework, type III sum of squares is utilized to control for the variance due to the other factors and covariates; however, this is not necessarily true in the likelihood ratio framework used here. If you disagree, please provide some reference. I think it would be much more convincing to show that there is still a significant variety x site interaction in first year survival or survival/number of flowers using the residuals from a model examining the correlation between size at planting and survival.

Validity of the findings

The clearest issue with this study is the lack of a refresher generation between seed collection and the experimental gardens. The observed differences in fitness could partly or fully be due to differential maternal effects rather than maladaptation. Presumably the temperatures were higher during the collection year for the southern variety compared to the northern variety, and therefore the southern variety may have been more primed to perform in warmer climates – exactly the pattern shown. This is briefly mentioned in the discussion, but it should be brought up in the methods as a caveat and the implications should be better explained in the discussion. An additional point of interest that could be included in the discussion on priming (~Line 498) would be the conditions during the year where seed was collected. What was the mean annual temp that year for each site and how did it relate to the growing season conditions?

In the abstract (L39-41), the conclusion that both populations are maladapted to their home environment. I would respectfully disagree – the southern variety has higher relative fitness in their how environment, and nothing is said about absolute fitness (see point above). Just because a population has higher fitness in a different environment does not mean that it is maladapted. For instance, in Clausen, Keck and Heisey’s seminal work, their high elevation pops performed better at low sites than high sites, but were the only population to survive at high sites. You provide a similar definition on L82. If you want to define maladaptation by home:away criteria, it is should written out explicitly as this is not the norm.

Additional comments

A key property of an adaptation lag is that weather during the experiment is more closely matched to historical climate of the distant variety rather than the native population. The data here perfectly matches this expectation, and I think it would be nice to lay this prediction out with the other prediction of adaptation lag, possibly on Line 136-137 or in the discussion. A related comment, Line 234 has the results that correspond to this prediction, but are in the methods.

I think a management-oriented person may be interested in absolute levels of fitness in this study – that is, did either the southern or the northern variety produce enough offspring to replace themselves. I know seed data was not collected, but any citations or observations in the discussion would be useful information.

Line 123: My personal opinion: The equation here seems unnecessary and breaks the flow of the intro.

Line 150: Any idea how old these perennials can exist in natural populations? Does your design encompass the typically lifespan or the most important part of it? Not necessary to include if unknown, but the reader will be wondering it here.

Line 155: Don’t need the parenthetical phrase as these were defined in the paragraph above.

Line 195: surveyed weekly, monthly, once a year?

Line 208 (and below): please add versions and citations for each package

Line 367: models 40 and 60? Are these defined somewhere.

Line 431: Does the structure of this low level of genetic diversity suggest that there has been a bottleneck or founder event in either population that caused maladaptation? Or are there similarly low levels of diversity in each population?

Figure 1: hard to tell light gray from dark gray points. How about using shapes?

Figure 3: Pretty sure the labels are wrong here as these temps did not match Fig 2 and the most northern garden did not have the highest mean annual temperature. Figure S6 has the same problem.

Figure 5: what do error bars represent (95% CI?, standard error around mean?). Please include in the legend

Reviewer 2 ·

Basic reporting

The introduction cites relevant literature, and generally provides the background necessary to understand the experiment.
The grammar is mostly good, but there are a few small errors. I note those that I caught in my minor comments.
The raw data is on github and the metadata is clear. The climate data is not included, as far as I could tell.
The figure captions are very clear and the figures are nice. A few may be unnecessary (see comments below).

1. The quotation of Kawecki and Ebert (line 82) seems misplaced, since it's a definition of a metric of local adaptation, but local adaptation has been discussed earlier in the paragraph too. Maybe provide more context for this quote, or use your own phrasing.
2. line 114: local adaptation is not mutually exclusive with insensitivity to temperature, as populations could be locally adapted to factors other than temperature.
3. line 137: this sentence is missing words, so its meaning is not clear. Regardless, it may be a good idea to be conservative about what you say you can disentangle, as a transplant with two populations will always have many confounding factors.
4. Fig 1c: I don't think you can technically assess local adaptation at the level of subspecies without a comparison to the other subspecies. This panel might be better described as niche limitation of subspecies, with no local adaptation.
5. Fig 2: I don't think you need to show the warmest quarter plot because you never discuss it in the text.
6. I think that Fig 3 could be moved to the supplement, and I don't think the warmest quarter plot is necessary because it is not discussed in the text.
7. Fig 5: The legend text is too small.
7. Fig S2 is unnecessary.
8. The contents of Figs S5 and S6 are not ever discussed in the text, perhaps consider omitting these figures.
9. I had to zoom in quite a bit to read the text on figures S4-S7, but the figures are high enough quality that they are legible once you zoom in.
10. The information in Table S3 would be more clear if it were presented in a plot.
11. The caption of Fig S7 says "raw data" but these are means (e.g. mean size) so that term isn't correct.

Experimental design

Understanding plant responses to climate change is an important current topic, and transplant experiments are a very powerful way of measuring potential responses. This paper contains multiple years of performance data from transplant gardens, and combines a reciprocal transplant with an over-the-edge transplant. These techniques are labor intensive and as a result, they are relatively rare, so this case study is a worthy contribution to the scientific literature.
The methods are generally appropriate, and described clearly with plenty of detail.
1. However, please give more information near line 293 on how these three metrics of local adaptation provide information about habitat quality, variety quality, and fit to environment. What can you learn from each metric of local adaptation that is not revealed by the others?
2. Some of the description of climate conditions in the methods (paragraph beginning 234) might be better placed in the results.
3. line 239: I'm not sure the far southern gardens historic means are relevant, as they were not experienced by plants during the experiment, nor are they the conditions these populations adapted to.
4. line 258: Did you consider trying a quadratic term for temperature? Is it possible that there might be an optimal temperature with declines above and below it? Or are you assuming a priori that all the experimental temperatures are above these optima, and that there will be linear relationships only?
6. line 306: Why did you choose not to include the beyond the edge gardens as "away" sites for both varieties? Explain in the text.

Validity of the findings

1. Can you evaluate the relative importance of survival vs. flowering on average performance for these varieties? Which life-history stages pose the largest challenge?
2. In the methods, you state that you measured when plants flower. Does their phenology give you any more clues about how temperature is affecting performance?
3. Paragraph beginning line 419: One issue with this experiment is that with just two populations, it is hard to have confidence in the magnitude of the temperature effect relative to other garden effects and intrinsic qualities of the populations. For example, could the results not also be explained by a higher garden quality at Svanvik, and greater load in the northern variety, without an effect of temperature? I don't mean to say that temperature is implausible--I think it's likely important, but it's hard to disentangle its effects from other factors with this design. The two populations used appear to differ by about 1 degree in their historic normal MAT. Is this difference enough to explain the huge difference in performance between the two varieties shown in figure 5b, where the red population's performance does not fall as low as the blue population's maximum unless it experiences a temperature 5.5 degrees warmer than it's historic normal? If you re-made your temperature model, and instead of actual garden temperatures, used the deviation of garden conditions from each population's historic normal (i.e., temperature mismatch), do you still see the southern population outperforming the northern one? Figure 5b makes me think that you would.
4. Section on confounding effects: I think the biggest limitation of your study is that you do not really have replication within varieties. Even though you sample multiple sites for seed collection, your approach doesn't really treat these as independent replicates. Discuss how this limits you ability to isolate temperature from other factors that may differ between northern and southern habitats.
5. line 542: why are sites beyond the range necessary for consideration of maladaptation? couldn't maladaptation also be revealed by within-range transplants?
5. line 454: I don't think the far southern gardens historic means are relevant, as they were not experienced by plants during the experiment, nor are they the conditions these populations adapted to.
6. line 334: I think you mean Fig 4 not Fig 3.
7. line 345: "raw data" isn't appropriate here.
8. line 377: Fig 1a is the map, do you mean a different figure?
9. line 412: Do you mean local adaptation to historic temperatures?

Additional comments

56: responseS
60: perhaps "tolerance of" rather than "plasticity towards"
99: maintenance instead of maintaining
104: what do you mean by "fixed" here?
136: translocatioN
225: with instead of among
Use of Fig., Fig, and Figure are inconsistent.
Check that figure and table references actually reference the correct figure/table.
Check formatting of references (i.e., sometimes full names are used in main text).

·

Basic reporting

Language is basically sound. On line 412 “evidenced” is used as a verb, which is not really appropriate. “shown” or “revealed” will do just fine.

The literature review omits much work on reciprocal transplants and local adaptation from North America, including the classic studies of Clausen, Keck, and Hiesey. Many of these studies have involved Arctic species such as that of Mooney and Billings(1961), McGraw on Dryas octopetala, Chapin on Carex aquatilis, and Bjorkman et al. (2016). The authors should also be aware of our studies of Eriophorum vaginatum in a reciprocal transplant experiment, for which I list some references below. The starred references are particularly relevant

Clausen, J., Keck, D.D., and Hiesey, W.M. 1948. Experimental studies on the nature of species. III. Environmental responses of climatic races of Achillea. Carnegie Institute of Washington, Washington, DC.
Mooney, H.A., and Billings, W.D. 1961. Comparative physiological ecology of arctic and alpine populations of Oxyria digyna. Ecol. Monogr. 31: 1–29.
Bjorkman, A.D., Vellend, M., Frei, E.R., and Henry, G.H.R. 2016. Climate adaptation is not enough: warming does not facilitate success of southern tundra plant populations in the high Arctic. Glob. Chang. Biol. 23: 1540–1551. doi:10.1111/gcb.13417.
Souther, S., and McGraw, J.B. 2011. Evidence of Local Adaptation in the Demographic Response of American Ginseng to Interannual Temperature Variation. Conserv. Biol. 25: 922–931. doi:10.1111/j.1523-1739.2011.01695.x.
McGraw, J.B., and Antonovics, J. 1983. Experimental ecology of Dryas octapetala ecotypes: II. A demographic model of growth, branching, and fecundity. J. Ecol. 71: 899–912. Also see other papers in this series.
Bennington, C.C., and McGraw, J.B. 1995. Natural selection and ecotypic differentiation in Impatiens pallida. Ecol. Monogr. 65: 303–323.
Chapin III, F.S., and Chapin, M.C. 1981. Ecotypic differentiation of growth processes in Carex aquatilis along latitudinal and local gradients. Ecology 62: 1000–1009.
Curasi, S.R., Parker, T. C., Rocha, A.V., Moody, M.L., J. Tang, and N. Fetcher. 2019. Differential responses of ecotypes to climate in a ubiquitous arctic sedge: implications for future ecosystem C cycling. New Phytologist 223: 180-192.
*Schedlbauer, J. L., N. Fetcher, K. Hood, M. L. Moody, and J. Tang. 2018. Effect of growth temperature on photosynthetic capacity and respiration in three ecotypes of Eriophorum vaginatum. Ecology and Evolution 8: 3711-3725 doi: 10.1002/ece3.3939
Parker, T. C., J. Tang, M. B. Clark, M. M. Moody, and N. Fetcher. 2017. Ecotypic differences in the phenology of the tundra species Eriophorum vaginatum reflect sites of origin. Ecology and Evolution 7: 9775-9786. doi: 10.1002/ece3.3445
Chandler, J. L., J. B. McGraw, C. Bennington, G. R. Shaver, M. C. Vavrek, and N. Fetcher. 2015. Tiller population dynamics of reciprocally transplanted Eriophorum vaginatum L. ecotypes in a changing climate. Population Ecology 57:117-126. DOI 10.1007/s10144-014-0459-9
*McGraw, J. B., J. B. Turner, S. Souther, C. C. Bennington, M. C. Vavrek, G. R.Shaver, and N. Fetcher. 2015. Northward displacement of optimal climate conditions for ecotypes of Eriophorum vaginatum L. across a latitudinal gradient in Alaska. Global Change Biology 21: 3827–3835. doi: 10.1111/gcb.12991
Souther, S., N. Fetcher, Z. Fowler, G. R. Shaver, and J. B. McGraw. 2014. Ecotypic differentiation in photosynthesis and growth of Eriophorum vaginatum along a latitudinal gradient in the Arctic tundra. Botany 92: 551–561 dx.doi.org/10.1139/cjb-2013-0320
*Bennington, C. C., N. Fetcher, M. C. Vavrek, G. R. Shaver, K. J. Cummings, and J. B. McGraw. 2012. Home site advantage in two long-lived arctic plant species: results from two 30-year reciprocal transplant studies. Journal of Ecology 100: 841-851. doi: 10.1111/j.1365-2745.2012.01984.x
Fetcher, N., and G. R. Shaver. 1990. Environmental sensitivity of ecotypes as a potential influence on primary productivity. American Naturalist 136:126-131.
Shaver, G. R., N. Fetcher, and F. S. Chapin III. 1986. Growth and flowering in Eriophorum vaginatum: Annual and latitudinal variation. Ecology 67:1524-1535.

That said, the ms. contains many useful references of which I was unaware, so I hope this has been a helpful exchange of information.

Figures 3 and S5.
Is the legend reversed? It seems that Svanvik should be cold and Tartu should be the warmest site.

Raw data are available. Permits are provided.

Experimental design

The study is a welcome addition to our understanding of plant responses to climate change. Given the wide distribution of Primula nutans, it would be interesting to see if it could be replicated elsewhere in the circumpolar region.

The study was very carefully done, and the analyses seem to be appropriate. I am somewhat concerned about the use of flower number as a component of fitness. Ideally, seed production would be the best measure of fitness, but it was not available in this experiment. Is there any evidence for a correlation between seed production and number of flowers? One can easily envision a scenario where a plant produces many flowers, but they don’t become pollinated or the seeds abort. Perhaps the aster model should be developed for the survival and flowered or not performance measures only and the results compared to those for the complete model.

Validity of the findings

Discussion is appropriate in relation to the issues raised in the introduction. It could be improved by reference to the literature mentioned above as well as other studies. In particular, I would be interested to know whether other studies have reported such a dramatic improvement for southern varieties when moved north. Perhaps more space could be devoted to this topic and less to the confounding effects.

Additional comments

There is no need to cite all of the references listed above. I included them to make the authors more aware of some of the work that has been done elsewhere.

---

## Round 0.2 · Minor Revisions

· Academic Editor

Minor Revisions

Both the reviewers and I believe that you have addressed the comments sufficiently and that only minor revisions is needed.

Reviewer 1 ·

Basic reporting

The authors have a done an excellent job addressing my previous concerns (I was reviewer 1 for the previous draft). I have a few general comments below that reflect small edits and/or additions.

Experimental design

The authors have added sufficient detail to the methods to meet my previous concerns and do a good job of justifying the decisions that they made.

Validity of the findings

The results justify the conclusions and appropriate caveats are described in the results.

Additional comments

Line 205: I was a bit confused by the wording here - the photoperiod was set to 12hr days?

Line 221 & 223: Probably don't need the italics here. I would save it for your variables below (personal opinion)

Line 414: I am not sure what you mean by "The northern variety, however, did not experience historical temperature conditions in any of the experimental sites". Does this mean that the northern variety has experienced hotter temperatures in its native range at some point in the past than it experienced in the most southern site? The subsequent comparison with the coldest site makes it confusing.

Line 455: I think it would be useful to have a sentence here that says that the southern plants were no larger at planting than the northern plants. This statement was it the response to reviewers, but would be useful in the manuscript.

Line 490: I don’t actually think your results are in complete concordance with the previous photoperiod study. Your results show that plants that did survive in the southern gardens were able to flower (thus not necessarily needing the cold nights of the northern garden). Making a more nuanced statement here would be useful.

Line 539: I think this section was a useful addition. I also think that it could benefit from one more sentence that provides an explanation for what maladaptation due to drift or stochastic processes would like spatially (i.e. the degree of maladaptation would be random with respect to geographic locality).

Reviewer: Nic Kooyers

Reviewer 2 ·

Basic reporting

The authors have responded to reviewers' suggestions and edited the manuscript accordingly. I think that overall the manuscript has improved and is a better presentation of a careful and laborious experiment.

There are still some grammatical and spelling errors that could be corrected, particularly in the revised sections.

The authors should also check their references carefully, e.g., I couldn't find Yien et al. 2019 (line 311 in tracked changes version) in the references list. I didn't check other references, I just happened to notice that one because it sounds interesting and I wanted to look it up!

Experimental design

The methods section is clear and well written.

The aims of the study and the distinct value of within-range vs. beyond-range transplants are highlighted by edits the authors made to the introduction.

I have one follow up from a comment on the previous version (Reviewer 2, comment 2.20). I really appreciate the figure you provided in the response letter plotting performance against temperature deviation from the varieties' historic normals. This figure clearly shows that to an extent, the poor performance of the northern variety is not only explained by temperature (if it were, you would expect both to overlap once temperature deviations were accounted for). This could be stronger genetic load in the northern variety due to drift or expansion, or garden conditions that for some unknown reason favored the southern variety. But very importantly, temperature conditions during the experiment do not fully account for the performance differences between varieties. I think this deserves a mention in the discussion and the figure should perhaps be added to the supplementary materials.

Validity of the findings

I take some issue with the statement "As far as we are aware, this represents the strongest example of climate change induced maladaptation." (lines 823-824) I think you mean that the distance between the southern source and the test site where it performed best is the furthest, but intermediate distances to the north were not sampled, so you don't really know that the Svanvik site is the optimum for that population, it's just the best among those you chose. There are some really excellent, more highly replicated studies of adaptation lag at similar geographic scales (Wilczek et al. 2014, Bontrager and Angert 2019) , so consider being more specific about what aspect of yours is the strongest, should you decide that it's necessary to make that claim at all.

·

Basic reporting

No comment

Experimental design

No comment

Validity of the findings

On reading the comments about the water table at the collection sites and visiting some of them via Google Earth, I was struck by the possibility that the plants could be adapted to soils with higher salt content than that encountered in the experimental gardens. Could this have had any effect on the results?

Additional comments

McGraw et al 2015 use the phrase "adaptational lag" instead of adaptation lag. This phrase is also used in Aitken SN, Yeaman S, Holliday JA, Wang T, Curtis-McLane S (2008) Adaptation, migration or extirpation: climate change outcomes for tree populations. Evolutionary Applications, 1(1):95–111. https://doi.org/10.1111/j.1752-4571.2007.00013.xI, which is cited in McGraw et al. I suggest that the authors continue to use "adaptational lag" to avoid introducing multiple terms for the same phenomenon.

---

## Round 0.3 · accepted · Accept

· Academic Editor

Accept

The authors have sufficiently addressed the comments by the reviewers and I am satisfied with the current version.